# The Double-Edged Sword Effect of Generative AI Adoption on Students’ Sustainable Entrepreneurship Intentions

**DOI:** 10.3390/bs15121705

**Published:** 2025-12-09

**Authors:** Weiwei Kong, Haiqing Hu, Zhaoqun Wang, Jianqi Qiao, Jianjun Liu

**Affiliations:** School of Economics and Management, Xi’an University of Technology, Xi’an 710054, China; 1210512016@stu.xaut.edu.cn (W.K.); wangzhaoqun@xaut.edu.cn (Z.W.); 105623@xaut.edu.cn (J.Q.); 2212527468@stu.xaut.edu.cn (J.L.)

**Keywords:** generative AI adoption, sustainable entrepreneurial intentions, sustainable entrepreneurial self-efficacy, sustainable entrepreneurial fear of failure, artificial intelligence literacy

## Abstract

Grounded in regulatory focus theory, this study investigates the double-edged sword effect of generative AI adoption on sustainable entrepreneurial intentions and its underlying mechanisms. A questionnaire-based survey was conducted among 357 business students from public universities in China. The results reveal that generative AI adoption exerts a double-edged effect: it enhances sustainable entrepreneurial intentions by strengthening sustainable entrepreneurial self-efficacy through a promotion-focused pathway, while simultaneously undermining such intentions by heightening sustainable entrepreneurial fear of failure via a prevention-focused pathway. Moreover, artificial intelligence literacy moderates these relationships, amplifying the positive influence of generative AI adoption on entrepreneurial self-efficacy and attenuating its negative effect on fear of failure. This study enhances understanding of sustainable entrepreneurship amid the rise in generative AI, extends regulatory focus theory, and informs the development of AI-integrated sustainability education in academic institutions.

## 1. Introduction

Amid escalating global issues such as environmental degradation, resource depletion, and social inequality, sustainable entrepreneurship has emerged as a key pathway to simultaneously advance social equity, economic growth, and environmental sustainability, as it emphasizes problem prevention at the source rather than post hoc remediation ([23]; [32]). However, sustainable entrepreneurial behavior does not arise by chance; it is often shaped by individuals’ motivations and cognitive readiness developed in the initial phases of entrepreneurship ([3]; [45]). To better understand the psychological mechanisms that shape responsible entrepreneurial behavior, researchers have increasingly turned their attention to the notion of sustainable entrepreneurial intentions (SEI) ([46]). Defined as individuals’ intentional pursuit of ventures aligning profitability with environmental care and social value creation, these intentions act as a key cognitive driver of sustainable entrepreneurial behavior ([57]). Gaining insight into the drivers behind SEI is vital not only for shaping entrepreneurs committed to social and environmental values but also for contributing meaningfully to several UN Sustainable Development Goals, such as Goals 4, 8, 9, and 12 ([39]; [50]).

As generative AI technologies like ChatGPT and DeepSeek become increasingly embedded in educational settings and entrepreneurial activities, growing scholarly interest has emerged around how their adoption influences entrepreneurial intentions and behaviors ([4]; [18]; [65]). Compared to traditional AI tools, generative AI possesses capabilities such as open interaction with users, content generation, and simulated reasoning, evolving the relationship between entrepreneurs and AI from mere “tool use” to “intelligent collaboration” ([35]; [49]). This form of collaboration transforms entrepreneurs’ thinking patterns and behaviors, enabling them to identify more entrepreneurial opportunities and thereby stimulating stronger entrepreneurial motivation. However, prior research has predominantly focused on how conventional AI technologies are utilized within sustainable entrepreneurship, while discussions concerning generative AI and SEI remain scarce and mostly highlight its beneficial outcomes (see Table 1). In fact, as an emerging technology characterized by high uncertainty, generative AI may also bring about “dark sides” such as anxiety, increased cognitive load, and technological dependence during its use ([14]; [38]; [48]; [55]; [62]). The evidence points to a nuanced relationship between generative AI and entrepreneurial intentions, where its impact is not solely beneficial but also includes potential drawbacks, underscoring its paradoxical role as both an enabler and an inhibitor.

Current research predominantly employs frameworks such as Stimulus-Organism-Response (SOR), Entrepreneurial Event Model (EEM), Antecedents-Behaviour-Consequences (ABC), or Theory of Planned Behavior (TPB) models, focusing on how external technological stimuli influence individual entrepreneurial behavior, but provides insufficient explanation of the dynamic evolution of internal psychological mechanisms. Therefore, there is an urgent need to adopt more nuanced psychological theoretical perspectives that reflect the dynamic process of internal motivational conflicts and psychological trade-offs exhibited by individuals when faced with the adoption of emerging technologies. Regulatory focus theory (RFT) is adopted in this research to provide a theoretical explanation for the opposing psychological effects observed in the context of generative AI adoption (GAA) ([25]). The theory posits that individuals typically exhibit two motivational orientations when pursuing goals. Individuals driven by a promotion focus are oriented toward potential gains and often pursue risk-taking approaches to attain success ([41]), while those with a prevention focus are more attuned to potential losses and typically adopt cautious strategies to avert failure ([10]). Consequently, individuals with a promotion focus tend to view generative AI as an enabling resource for enhancing their competencies, thereby strengthening their confidence in developing sustainable entrepreneurial self-efficacy (SES) ([6]). Conversely, individuals exhibiting a prevention focus tend to be more sensitive to the risks and uncertainties associated with generative AI, which may heighten their sustainable entrepreneurial fear of failure (SEFF) ([11]). These two variables represent the positive and negative psychological pathways triggered by GAA, and they may play a mediating role in shaping SEI.

Furthermore, RFT asserts that individuals’ goal pursuit is significantly shaped by their personal experience and capability levels ([31]). However, prior studies have mainly highlighted how contextual elements—such as pressure from technological change, institutional backing, and environmental dynamics—moderate the outcomes of interest ([18]), with insufficient attention paid to entrepreneurs’ intrinsic cognitive abilities. With AI tools being increasingly embedded in educational and entrepreneurial contexts, artificial intelligence literacy (AIL)—referring to the ability to understand, evaluate, and effectively apply such technologies ([42]; [43]; [58])—has become a key competence shaping how successfully AI is adopted. Therefore, incorporating AIL into the theoretical framework is necessary to explore its moderating role in the dual psychological pathways through which GAA affects SEI, thereby uncovering the individual heterogeneity mechanisms underlying technology impact.

Grounded in the foregoing analysis, the present investigation utilizes RFT to uncover the complex, dual-natured impact of GAA on SEI. The empirical evidence, gathered from a cohort of 357 Chinese public university business majors, enriches the current academic understanding in several important respects. First, although traditional AI’s application and effects in sustainable entrepreneurship have been extensively studied, there is a noticeable scarcity of research dedicated to generative AI, which mostly concentrates on its advantageous impacts on SEI. This paper highlights the potential “negative psychological effects” triggered by GAA, demonstrating that GAA influences SEI through both positive facilitative and negative inhibitory pathways. Second, compared with existing studies that explore GAA and sustainable entrepreneurship using frameworks such as SOR, EEM, ABC, or TPB models, this study adopts an RFT perspective to capture the dynamic process of intrinsic motivational conflicts and psychological trade-offs experienced by individuals when adopting emerging technologies. Finally, AIL serves to moderate the double-edged sword effect that GAA exerts on SEI, effectively expanding the scope of conditions under which GAA influences SEI.

## 2. Theory and Hypotheses

### 2.1. Regulatory Focus Theory

Although prior studies have highlighted the effects of AI adoption on entrepreneurial intention, most focus on linear or uniformly positive pathways, leaving unexplored the dual psychological mechanisms suggested by RFT. According to RFT, individuals pursue goals through two distinct motivational orientations: promotion focus and prevention focus ([25]). A promotion focus is oriented toward growth, advancement, and the pursuit of gains, leading individuals to approach opportunities and take proactive actions to realize desired outcomes. In contrast, a prevention focus emphasizes security, responsibility, and the avoidance of losses, prompting individuals to adopt a cautious and risk-averse stance.

When applied to the context of generative AI adoption, these two orientations give rise to distinct cognitive evaluations and emotional reactions that influence sustainable entrepreneurial intention. Promotion-focused entrepreneurs tend to view generative AI as a means of opportunity exploration and capability enhancement. Such perceptions strengthen their SES—reflecting confidence in their ability to achieve sustainable goals—and consequently foster stronger SEI ([6]). Conversely, prevention-focused entrepreneurs are more sensitive to the risks, uncertainties, and ethical challenges of AI technologies. This cautious interpretation can increase SEFF, thereby reducing their SEI ([11]). Therefore, SES and SEFF represent two distinct psychological pathways aligned with promotion and prevention focuses, providing a theoretical rationale for their role as mediators in the proposed model rather than arbitrarily selected constructs.

Moreover, RFT posits that individuals’ goal pursuit processes are significantly influenced by their prior experiences and capability levels ([31]). In this context, AIL represents a composite variable capturing entrepreneurs’ technological knowledge, operational skills, and critical thinking. It is theoretically justified as a moderating variable because, according to RFT, individual capability levels shape how motivational orientations translate into psychological responses. Specifically, promotion-focused entrepreneurs demonstrating elevated AIL tend to view technological advancements positively, which relates to higher SES ([13]). Meanwhile, those oriented toward prevention are better equipped—thanks to greater AIL—to evaluate potential risks more objectively, which may reduce SEFF ([64]). Therefore, AIL moderates the strength of the association between GAA and the dual psychological pathways, rather than acting as an antecedent or mediator. To visually represent this theoretical logic, a conceptual model is proposed (see Figure 1).

### 2.2. The Mediating Role of SES

According to RFT, people exhibiting a promotion focus are inclined to concentrate on opportunities for growth, improvement of skills, and routes leading to success within their surroundings ([25]). When faced with the introduction of generative AI—a cutting-edge and rapidly evolving technology—such individuals tend to view it as an instrument for resource acquisition, cognitive enhancement, and the fulfillment of personal or business objectives. This perspective suggests that promotion-focused entrepreneurs may experience higher SES, reflecting their perceived ability to manage sustainable entrepreneurial tasks, which is theoretically expected to be associated with stronger SEI ([6]).

GAA may contribute to enhancing individuals’ perceptions of their capabilities in sustainable entrepreneurship from multiple dimensions ([5]). First, GAA assists entrepreneurs in performing critical tasks such as designing sustainable business models, interpreting environmental policies, and evaluating social impact ([60]). These activities may provide mastery experiences that reinforce perceived control over sustainability-related practices. Second, by enabling access to best practices and case-based learning, GAA offers vicarious experiences, allowing entrepreneurs to potentially draw confidence from observing others’ successes ([9]). Third, the intelligent feedback and strategic recommendations provided by generative AI could act as verbal persuasion, encouraging entrepreneurs to persist in exploring sustainable innovation and efficient resource utilization pathways ([19]). Finally, GAA might help reduce the complexity and uncertainty inherent in sustainable entrepreneurship, helping entrepreneurs manage emotional arousal by alleviating stress associated with environmental compliance and social impact management ([52]). Therefore, we propose the following:

**H1a.** 
*GAA will positively influence SES.*


Numerous studies have shown that entrepreneurial self-efficacy—the confidence individuals have in their entrepreneurial abilities—significantly influences their entrepreneurial intentions ([20]; [21]; [29]; [30]; [34]; [36]).

SES is theoretically expected to enhance SEI by strengthening individuals’ perceptions of goal feasibility, stimulating intrinsic motivation, and regulating emotional responses ([3]; [6]). Entrepreneurs who perceive high self-efficacy may demonstrate greater confidence in executing essential activities, including the adoption of eco-friendly technologies, efficient resource utilization, adherence to environmental standards, and effective management of social responsibilities. Consequently, they may be more likely to develop clear, stable, and enduring SEI ([37]; [47]). Such individuals view sustainable entrepreneurship as both an aspirational goal and a feasible path, displaying enhanced motivation and behavioral engagement across cognitive, emotional, and behavioral dimensions. Furthermore, sustainable entrepreneurial activities are frequently motivated by a deep commitment to social responsibility and a clear sense of purpose toward societal well-being ([7]). People possessing strong self-efficacy may not only have confidence in their ability to act but also feel a moral obligation to act, which further strengthens the ethical and emotional basis of entrepreneurial motivation. Therefore, we propose the following:

**H1b.** 
*SES will positively influence SEI.*


**H1.** 
*SES could act as a mediator in the relationship between GAA and SEI.*


### 2.3. The Mediating Role of SEFF

RFT suggests that those primarily guided by a prevention focus prioritize processing information through the lens of minimizing risks and ensuring security, placing particular importance on fulfilling obligations and avoiding mistakes ([10]). In the context of GAA—a complex and rapidly evolving technological stimulus—prevention-focused entrepreneurs may perceive higher potential for failure and loss, which is theoretically associated with increased SEFF and, in turn, could be linked to lower levels of SEI.

Firstly, the rapid iteration and high technical complexity of generative AI impose significant learning barriers and cognitive burdens. Entrepreneurs oriented toward prevention often worry about their limited proficiency in data literacy and technical skills necessary to efficiently understand and apply AI technologies; this gap in expertise may underlie their elevated SEFF ([51]). Secondly, the reliance of generative AI on data and algorithms in entrepreneurial decision-making introduces risks related to resource allocation and algorithmic biases, which could increase apprehensions about failures resulting from erroneous decisions or resource mismanagement ([15]). Furthermore, generative AI introduces a range of ethical and regulatory challenges—including potential breaches of data protection, algorithmic unfairness, and legal accountability issues ([56])—which can heighten entrepreneurs’ worries about reputational harm and legal repercussions, thereby amplifying their fear of failure. Finally, the complexity and volume of generative AI information and feedback could cause cognitive overload ([63]), potentially weakening individuals’ decision-making confidence and increasing their anxiety about potential failure. Therefore, we propose the following:

**H2a.** 
*GAA will positively influence SEFF.*


Prior research has consistently shown that elevated fear of failure in entrepreneurship substantially suppresses individuals’ intentions to pursue entrepreneurial activities. Fear of entrepreneurial failure encompasses concerns about others’ expectations (such as the fear of failing to meet the hopes of society or close acquaintances), anxiety over insufficient funding, doubts regarding the value of the business idea, worries about the time commitment involved, and questions about one’s managerial capabilities ([11]; [21]). These concerns tend to be particularly pronounced within the complex context of sustainable entrepreneurship.

Specifically, fear of failing to meet the expectations of society or close acquaintances creates intense external evaluative pressure on entrepreneurs, leading to more conservative decision-making ([12]). Concerns about insufficient funding can undermine confidence in project sustainability, prompting them to abandon or scale down entrepreneurial plans when resources are limited. Doubts regarding the value of the business idea may reduce identification with sustainable entrepreneurship and lower motivation for sustained effort. Worries about time investment force entrepreneurs to balance personal life and entrepreneurial tasks, diminishing their commitment to long-term and complex entrepreneurial endeavors. Additionally, doubts about managerial capabilities can exacerbate failure anxiety, negatively influencing confidence in coping with multiple challenges. Collectively, these multidimensional SEFF may trigger avoidance and defensive behaviors, thereby potentially suppressing proactive exploration, innovation, and sustained engagement ([16]). Such risk-averse attitudes not only weaken the long-term commitment and resilience necessary for entrepreneurship but also significantly reduce the likelihood of entrepreneurs forming and maintaining SEI. Therefore, we propose the following:

**H2b.** 
*SEFF will negatively influence SEI.*


**H2.** 
*SEFF could act as a mediator in the relationship between GAA and SEI.*


Combining H1 and H2, we further propose the following:

**H3.** 
*SES and SEFF play dual mediating roles in the relationship between GAA and SEI.*


### 2.4. The Moderating Role of AIL

RFT posits that individuals’ goal pursuit is driven not only by intrinsic motivations—such as promotion or prevention focus—but also significantly influenced by personal experience and capability levels ([31]). AIL is conceptualized as a personal capability reflecting individuals’ knowledge, skills, and cognitive competence in understanding and applying AI technologies, and is theoretically expected to shape the strength of psychological responses to generative AI rather than directly causing them ([42]; [43]; [58]).

Those who exhibit a promotion focus generally pursue development, innovation, and success, actively embracing novel technologies to facilitate their objectives. This positive motivational orientation is likely to be associated with better behavioral outcomes when supported by corresponding capabilities. Enhanced AIL may strengthen individuals’ mastery over generative AI, thereby creating a high degree of alignment between motivation and ability ([42]). Specifically, those possessing advanced AIL can more effectively comprehend the underlying mechanisms, scope of functions, and appropriate contexts for applying generative AI, potentially facilitating more seamless and manageable positive user experiences. Such experiences not only raise the probability of successful task completion but also foster the development of mastery-related achievements, thereby strengthening individuals’ confidence in their sustainable entrepreneurial abilities ([13]; [43]). Conversely, entrepreneurs possessing limited AIL—although driven to engage with generative AI—often face challenges in understanding, navigating operational complexities, and accurately interpreting system feedback, primarily due to gaps in relevant expertise and technical proficiency. These challenges may lead to frustrating experiences that diminish the empowering effects of AI and could trigger cycles of dependence, anxiety, and ineffective attempts, which in turn may weaken their confidence in entrepreneurial abilities and weaken their SES ([26]; [42]). Therefore, we propose the following:

**H4a.** 
*AIL positively moderates the relationship between GAA and SES.*


Those exhibiting a prevention focus tend to emphasize possible uncertainties and risks of failure associated with generative AI—an intricate and rapidly advancing emerging technology—which may trigger increased SEFF ([64]). However, the intensity of this risk perception and fear experience is not solely determined by the technological characteristics but is significantly influenced by the individual’s understanding of and ability to manage AI. In other words, the level of AIL could affect an individual’s psychological resilience and coping strategies when confronted with AI-related challenges ([27]). Specifically, individuals with higher AIL tend to possess stronger information judgment and technical operation abilities when adopting generative AI. They may be able to identify and avoid algorithmic biases, data pitfalls, and compliance risks, thereby effectively reducing their perception of uncertainty ([13]; [43]). Having a perceived sense of mastery and the ability to foresee potential challenges can contribute to reducing anxiety related to failure, thereby diminishing the apprehension associated with using generative AI. In contrast, entrepreneurs with lower AIL are more prone to operational errors and misinterpretation of information when facing complex technologies, which could amplify their anticipation of potential failure. For them, GAA might become a trigger for fear ([33]). Therefore, we propose the following:

**H4b.** 
*AIL negatively moderates the relationship between GAA and SEFF.*


Therefore, considering H1 to H4b, we further propose the moderated mediation hypothesis:

**H5a.** 
*AIL moderates the mediating role of SES between GAA and SEI.*


**H5b.** 
*AIL moderates the mediating role of SEFF between GAA and SEI.*


## 3. Methods

### 3.1. Measures

To enhance the robustness of measurement, this study adopted well-established constructs from prior research, with appropriate modifications to suit the Chinese context. The scale items were first rendered into Chinese by bilingual experts and subsequently retranslated into English by an independent researcher to verify semantic equivalence. Any discrepancies between the two English versions were discussed and revised to ensure conceptual accuracy. Thereafter, minor wording refinements were made to improve cultural fit. A preliminary survey involving business students from multiple universities was then carried out to evaluate the clarity and contextual relevance of the items. All constructs were rated on a seven-point Likert-type scale (see Appendix A).

**Generative AI adoption (GAA):** This scale was adapted from [18] ([18]) and localized to reflect the specific AI tools and scenarios relevant to the Chinese context. Participants reported how often and to what degree they utilized generative AI tools within the context of sustainable entrepreneurship, and this construct was measured using five items. To enhance respondent comprehension and ensure conceptual equivalence across participants, examples of commonly used generative AI systems (e.g., ChatGPT, DeepSeek, ERNIE Bot, and DouBao) were included in the questionnaire. These examples were intended to anchor respondents’ understanding of generative AI in familiar, functionally comparable applications while minimizing ambiguity regarding what “AI tools” refer to. Importantly, the items themselves were phrased to capture platform-agnostic behaviors (e.g., frequency, breadth of use, and purpose), rather than platform-specific features, thereby operationalizing a general construct of GAA. This operational decision helped ensure measurement consistency and conceptual clarity among respondents who share similar AI use experiences.

**Sustainable entrepreneurial intentions (SEI):** This scale was adapted from the entrepreneurial intention scale by [34] ([34]), with revisions based on [6] ([6]) to reflect sustainability-oriented entrepreneurial goals. The scale includes five items that assess respondents’ intentions to pursue entrepreneurial ventures with a focus on sustainability.

**Sustainable entrepreneurial self-efficacy (SES):** The scale was modified from [1] ([1]), which originally measured social entrepreneurial self-efficacy. Items were adjusted to better reflect self-efficacy in the context of sustainable entrepreneurship, with three items included to capture respondents’ confidence in pursuing sustainable ventures.

**Sustainable entrepreneurial fear of failure (SEFF):** This scale is a simplified version of the multidimensional fear of failure scale developed and validated by [11] ([11]). While the original scale measures fear of failure in the general entrepreneurial context, this adapted version specifically addresses emotional and cognitive responses to fear of failure in sustainable entrepreneurship. It focuses on concerns about the consequences of failure in sustainability-focused ventures, feelings of shame, and psychological stress, with five items included.

**Artificial intelligence literacy (AIL):** This scale was adapted from [58] ([58]), which is designed to assess AIL among non-experts. Given its proven reliability and validity in academic settings ([43]; [59]), the scale was deemed suitable for this study. A screening question was included at the beginning of the survey to ensure that respondents had prior experience with generative AI tools. The final scale consists of 12 items, with items 2, 5, and 11 reverse-coded.

**Control variables:** Following previous studies ([17]; [40]), the study controlled for potential demographic and personal background factors that could influence SEI. The control variables include gender, age, degree, and business experience.

### 3.2. Sampling and Data Collection

To effectively control for potential common method bias arising from respondent homogeneity, this study employed a stratified sampling method to ensure representativeness at both regional and institutional levels. The sample was stratified based on two key factors: geographical region (eastern, central, and western China) and university type (research-oriented vs. teaching-oriented universities). Specifically, universities in the eastern, central, and western regions were selected to capture diverse economic and educational contexts. Further, universities were categorized into research-oriented and teaching-oriented institutions to reflect different academic focuses and available resources. This stratification approach ensured diverse representation of respondents from various academic and geographic backgrounds, enhancing the generalizability of the results.

The survey specifically targeted undergraduate and postgraduate business students enrolled at public universities across China. Business students were chosen as the study sample for two primary reasons: First, they constitute a group of prospective entrepreneurs. Second, they typically have greater access to emerging technologies like generative AI and receive corresponding training in sustainability-oriented entrepreneurship, which makes them particularly suitable for this study.

This study used purposive sampling to ensure that respondents had a basic understanding of artificial intelligence technologies and relevant entrepreneurial experience. A screening question was included at the beginning of the survey to confirm whether respondents had used generative AI tools (e.g., ChatGPT, DeepSeek, ERNIE Bot, DouBao). Only those who confirmed their experience with these AI tools were eligible to participate in the survey. Additionally, the survey included open-ended questions designed to assess respondents’ familiarity with AI concepts (e.g., their understanding of AI tools and specific examples of how they use AI tools in their daily academic and personal activities). Through this process, it was confirmed that participants demonstrated a foundational understanding of AIL.

Before the formal distribution of the questionnaire, a pilot test was conducted with 58 respondents to assess the clarity, reliability, and cultural appropriateness of the measurement scales. Based on the feedback from the pilot test, several items were slightly revised to improve their wording and ensure they were contextually relevant. For instance, terms related to AI tools and entrepreneurial concepts were clarified to ensure that participants could easily understand them in the context of Chinese culture and language.

The formal survey was conducted from January to April 2025. Questionnaires were distributed using both online platforms (such as wjx.cn and Credamo) and offline channels (including entrepreneurship courses and innovation labs). To ensure that respondents had actual experience with generative AI, a screening question was placed at the beginning of the questionnaire: “Have you ever used generative AI tools (e.g., ChatGPT, DeepSeek, ERNIE Bot, DouBao)?” Responses were automatically terminated if this criterion was not met.

A total of 520 questionnaires were distributed, and 398 responses were collected. After removing invalid and incomplete questionnaires, 357 valid responses remained, resulting in an effective response rate of 68.65%. Detailed demographic characteristics are presented in Table 2.

### 3.3. Non-Response and CMB

To address potential CMB, several diagnostic procedures were conducted. First, Harman’s single-factor test was applied while controlling for potential measurement errors. The five extracted factors jointly accounted for 64.321% of the total variance, and the largest individual factor explained 25.452%, which falls within acceptable thresholds. Second, as indicated in Table 3, the confirmatory factor analyses demonstrated that the five-factor configuration provided the best model representation, with CFI and TLI values exceeding 0.995 and RMSEA remaining well below 0.05. Third, an additional common latent factor was incorporated into the CFA model to compare its loadings against those of the original structure. Following the criterion of [54] ([54]), if the inclusion of this factor improved CFI and TLI by more than 0.1, substantial method bias would be inferred. Nevertheless, the comparative results revealed negligible variation in the model indices. Collectively, these findings indicate that CMB was not a serious concern in this study.

## 4. Results

### 4.1. Reliability and Validity

Before conducting reliability and validity analyses, reverse-coded items (Items 2, 5, and 11 in the AIL scale) were recoded to ensure consistency in scoring direction. Data normality was assessed using both the Kolmogorov–Smirnov (K–S) and Shapiro–Wilk (S–W) tests, as well as skewness and kurtosis statistics. The results of the K–S and S–W tests were non-significant (*p* > 0.05), and all skewness and kurtosis values fell within the acceptable threshold of ±2 ([28]), indicating that the data satisfied the normality assumption for subsequent parametric analyses.

To ensure methodological transparency, the measurement model was evaluated using multiple reliability and validity indicators. First, we used Cronbach’s Alpha (CA) as a measure of reliability. The CA and CR values for all five variables exceeded the threshold of 0.70 (see Table 4). Second, we assessed the validity of the scale. The AVEs for all five variables exceeded 0.5, and the square roots of the AVEs for each variable were greater than 0.70 (see Table 4). Additionally, the correlation coefficients between variables did not exceed the square root of the AVEs (see Table 5), and the HTMT values were less than 0.85 (see Table 6) ([24]). Finally, the VIF values were far below the critical value of 10, indicating that the regression model did not exhibit severe multicollinearity issues. In summary, these five variables demonstrated high reliability and validity, meeting the research standards.

### 4.2. Hypothesis Tests

Table 7 shows the results of the mediation effect regression analysis. The results indicate that GAA has a significant positive effect on SES (M2, β = 0.411, *p* < 0.001) and SEFF (M4, β = 0.175, *p* < 0.01). Therefore, hypotheses H1a and H2a are supported. In addition, GAA significantly influenced SEI (M6, β = 0.426, *p* < 0.001), which is a prerequisite for the mediation effect to be established in this study. SES significantly and positively influenced SEI (M7, β = 0.348, *p* < 0.001), while SEFF significantly and negatively influenced SEI (M8, β = −0.137, *p* < 0.001). Therefore, H1b and H2b were supported. When the model simultaneously considered GAA, SES, and SEFF, the coefficient of GAA’s influence on SEI was significantly reduced (M9, β = 0.306, *p* < 0.001). These results preliminarily support H1, H2, and H3.

Subsequently, we used the Bootstrap to further validate hypotheses H1, H2, and H3 ([44]). We selected 5000 repeated samples with a confidence level of 95%. The results are shown in Table 8. The indirect effect value of GAA on SEI through SES is 0.1432, with a 95% CI of [0.0856, 0.2059], which does not include 0. Therefore, the mediating effect of SES is significant, validating hypothesis H1. The indirect effect value of GAA on SEI through SEFF is −0.0241, with a 95% confidence interval of [−0.0526, −0.0032], which does not include 0. Therefore, the mediating effect of SEFF is significant, and H2 is verified. In addition, the total indirect effect coefficient of GAA on SEI through two mediating variables is positive, and the effect is significant (Coef. = 0.1736, 95% CI [0.1096, 0.2411]). The indirect effects of SES (Coef. = 0.1468, 95% CI [0.0864, 0.2113]) and SEFF (Coef. = −0.0268, 95% CI [−0.0553, −0.0051]) were also significant. Therefore, H3 is supported.

Table 9 shows the results of the regression analysis of the moderating effect. M12 added the interaction term between AIL and GAA to M11, and the coefficient of the interaction term was significantly positive (M12, β = 0.092, *p* < 0.01). M14 adds an interaction term between AIL and GAA to M13. The coefficient of the interaction term is significantly negative (M14, β = −0.228, *p* < 0.001). Thus, H4a and H4b are supported.

Table 10 presents the results of the moderated mediating effect test conducted using the Bootstrap. The effect of GAA on SEI through SES is significant at both high AIL levels (Coef. = 0.2043, 95% CI [0.1209, 0.2981]) and low AIL levels (Coef. = 0.1012, 95% CI [0.0283, 0.1710]), and the difference between groups is significant (Coef. = 0.1030, 95% CI [0.0133, 0.2242]). The index of moderated mediation is 0.0328, with a 95% CI of [0.0038, 0.0714], which does not include 0, thus H5a is supported. The effect of GAA on SEI through SEFF was significant at both high AIL levels (Coef. = 0.0299, 95% CI [0.0007, 0.0728]) and low AIL levels (Coef. = −0.0791, 95% CI [−0.1404, −0.0275]), and the difference between groups is significant (Coef. = 0.1090, 95% CI [0.0342, 0.1993]). The index of moderated mediation is 0.0347, 95% CI [0.0115, 0.0638], which does not include 0, thus H5b is supported.

To further interpret the moderating role of AIL, we compared the practical implications of high and low AIL levels based on the simple slope analysis (see Figure 2 and Figure 3). In real-world terms, individuals with high AIL are typically more capable of understanding the logic and limitations of generative AI tools, integrating them effectively into opportunity recognition, problem-solving, and innovation design. This allows them to experience a stronger sense of control and confidence in AI-supported entrepreneurial tasks, thereby enhancing SES and reducing SEFF.

Conversely, individuals with limited AIL frequently find it difficult to assess the credibility of AI-generated content and are inclined to follow algorithmic guidance passively. Consequently, they experience higher levels of uncertainty and anxiety when using generative AI for entrepreneurial purposes, which heightens their fear of failure and suppresses self-efficacy. These behavioral and psychological distinctions clarify how AIL moderates the relationship between GAA and both SES and SEFF, providing richer practical meaning for the statistically observed moderation effect.

### 4.3. Structural Equation Modeling (SEM) Analysis

To further enhance the robustness of the empirical evidence, we conducted a covariance-based SEM analysis to validate the overall research model. Compared with regression analysis, SEM allows simultaneous estimation of multiple dependent relationships and provides a more comprehensive assessment of model fit.

We used the SEM software AMOS 21.0 to test the hypothesized model, as shown in Figure 4. The model fit indicators were χ^2^/df = 1.092, *p* = 0.059, RMR = 0.127, CFI = 0.993, GFI = 0.914, AGFI = 0.899, RMSEA = 0.016, indicating that our model is acceptable ([61]). Specifically, GAA has a significant positive effect on SES (β = 0.489, *p* < 0.001) and SEFF (β = 0.264, *p* < 0.001), while SES positively affected SEI (β = 0.478, *p* < 0.001) and SEFF negatively affected SEI (β = −0.176, *p* < 0.001). These results support H1a, H1b, H2a, and H2b. Moreover, the moderating effects of AIL were also verified. The interaction term between GAA and AIL significantly enhanced SES (β = 0.091, *p* < 0.01) and reduced SEFF (β = −0.257, *p* < 0.001), supporting H4a and H4b. Control variables (gender, age, degree, and business experience) did not exhibit significant effects on SEI. Overall, the SEM results reinforce the robustness of the regression-based findings and provide additional empirical support for the proposed research model.

## 5. Discussions and Conclusions

Generative AI, as a disruptive enabling technology, is increasingly integrated into entrepreneurial education and practice, exerting significant influence on entrepreneurs’ motivation and psychological states. Grounded in RFT, this study constructs a dual-pathway framework to explore the ambivalent influence of GAA on individuals’ SEI and its underlying mechanisms. Drawing on data from a sample of business students at Chinese public universities, the empirical findings reveal that GAA positively impacts SEI by enhancing SES (promotion focus), while concurrently exerting a negative effect by heightening SEFF (prevention focus). Furthermore, AIL plays a critical moderating role, amplifying the positive influence of GAA on SES and attenuating its adverse impact on SEFF.

### 5.1. Theoretical Contributions

First, we broaden the research scope concerning the relationship between GAA and SEI. Existing literature predominantly centers on the application and impact of traditional AI within sustainable entrepreneurship, such as AI tool usage ([65]), integration of AI in education ([4]), and the facilitation of sustainable entrepreneurship through AI and big data analytics ([8]). However, research specifically addressing generative AI’s influence on SEI remains nascent. While preliminary findings suggest that GAA can enhance SEI by increasing perceived desirability and feasibility ([17]), these studies have primarily emphasized its positive effects and have yet to sufficiently investigate the potential psychological risks and adverse consequences—such as anxiety, cognitive overload, and heightened technology dependence—that may arise during actual usage. This paper uncovers that GAA impacts SEI through both promotive and inhibitory pathways, thereby challenging the predominantly rationalistic assumptions commonly found in AI adoption research. This dual-path finding not only responds to calls for a more nuanced and dialectical examination of GAA ([14]; [48]; [62]) but also offers enhanced theoretical insights into the complex psychological dynamics underlying AI use.

Second, we offer a dynamic explanatory framework grounded in RFT to elucidate the complex mechanisms through which GAA influences SEI. Existing research on generative AI and entrepreneurial behavior predominantly adopts theoretical frameworks such as SOR, EEM, ABC, or TPB, focusing mainly on external environmental stimuli or behavioral drivers and emphasizing how technology usage triggers entrepreneurial intentions or behaviors. However, these approaches often overlook the dynamic internal motivational conflicts and psychological trade-offs individuals experience when adopting emerging technologies. In contrast, RFT captures the dual psychological mechanisms activated when individuals face high-uncertainty technologies like generative AI: on one hand, entrepreneurs may experience a growth-oriented promotion focus driven by technological empowerment, enhancing their SES; on the other hand, risk perception and fear of failure may trigger a prevention focus, eliciting SEFF. This study approaches the interplay of these positive and negative pathways as a dynamic process, systematically identifying and empirically validating the coexisting psychological mechanisms of SES and SEFF. Consequently, it provides a more nuanced and psychologically rich theoretical foundation for understanding the formation and evolution of SEI in the context of GAA.

Finally, we highlight the moderating role of AIL in the mechanism through which GAA influences SEI, thereby expanding the boundary conditions of generative AI’s impact. Existing research on the effects of AI adoption on entrepreneurial behavior has predominantly focused on contextual moderators such as technological pressure, policy support, and external environmental factors ([18]), with relatively limited attention paid to individual cognitive abilities, especially differences in technological literacy among entrepreneurs. By introducing AIL as a key moderating variable, this study finds that it plays a significant role in both pathways through which GAA affects SEI: on one hand, higher AIL strengthens the positive effect of GAA on SES; on the other hand, it effectively mitigates the SEFF, thereby reducing the negative effect. This finding not only underscores the shaping influence of individual capabilities on technology outcomes but also responds to the recent trend of “individualization” in technology acceptance research. Accordingly, it offers a conceptual basis for interpreting the varying impacts of generative AI across individuals.

### 5.2. Managerial Implications

First, to enhance the positive pathway, it is essential to leverage the empowering potential of GAA to stimulate SEI. The data indicate that GAA has a significant positive effect on SES, suggesting that active engagement with generative AI tools can effectively strengthen individuals’ confidence in managing entrepreneurial tasks. For students, they can use AI tools to analyze market trends, generate creative ideas, and simulate decision-making for sustainable business opportunities. Such hands-on engagement can strengthen both their technological skills and self-efficacy in pursuing sustainable entrepreneurship. For educators, as key facilitators of entrepreneurship education, it is crucial to systematically integrate generative AI into entrepreneurship curricula and promote its pedagogical embedding. This can be achieved through project-based courses, AI-assisted entrepreneurial training, and interdisciplinary innovation workshops that provide practice-oriented empowerment scenarios. In particular, educators may design AI-supported entrepreneurial projects that encourage students to identify sustainability challenges and use AI tools for solution prototyping. Aligning such activities with the UN SDGs can further foster a generation of sustainability-oriented entrepreneurial talent.

Second, to address both the positive and negative psychological pathways of GAA, institutions should support students in enhancing SEE while managing SEFF. The results show that GAA can simultaneously increase SES and SEFF, indicating the need to develop both technical competence and emotional resilience. For students, this means not only actively applying generative AI to strengthen confidence in entrepreneurial tasks, but also cultivating strategies to cope with uncertainty and setbacks, such as reflecting on challenges and analyzing failed decisions. For educators, entrepreneurship education should adopt a dual-focus approach that integrates capability development with psychological intervention. Practically, universities can employ scenario-based learning that simulates entrepreneurial setbacks in safe and controlled environments. Educators can also invite entrepreneurs who have experienced failure to share reflective narratives, complemented by “Learning from Failure” workshops that help students emotionally process failure and reconstruct its developmental value. Additionally, introducing reflective journals, team-based debriefing sessions, and AI-assisted resilience training modules can support students in systematically identifying learning opportunities from failure and building adaptive coping mechanisms.

Finally, the moderating effect of AIL underscores its strategic importance for both individual and institutional development. The moderation analysis demonstrates that higher AIL amplifies the positive impact of GAA on SES while attenuating its effect on SEFF. For students, this means that continuous improvement in data literacy, algorithmic thinking, and ethical awareness is essential for managing the dual effects of GAA. For educators, AIL should be recognized as a core entrepreneurial competency. Universities may establish AIL labs and cross-disciplinary training modules that integrate entrepreneurship, technology ethics, and risk management. Simulation-based learning and AI-driven decision labs can also be used to immerse students in real-time entrepreneurial dilemmas, allowing them to practice balancing opportunity pursuit with risk prevention. These evidence-based initiatives equip students with the capacity to leverage AI tools efficiently, exercise sound judgment amid uncertainty, and build the adaptability required for sustainable entrepreneurship in the age of AI.

### 5.3. Limitations and Future Directions

First, the sample consisted primarily of business students enrolled in public universities across China. While this population provides a relevant context for examining SEI, the findings may not fully generalize to entrepreneurs from different cultural, educational, or occupational backgrounds. Subsequent studies may expand the sample coverage to encompass entrepreneurs and nascent founders from various regions and institutional contexts, thereby improving the generalizability of the findings.

Second, in operationalizing GAA, this study included illustrative examples of widely used text-based platforms such as ChatGPT, DeepSeek, ERNIE Bot, and DouBao. This design choice was intended to ensure respondent familiarity and conceptual clarity, given the prevalence of these tools in Chinese higher education and entrepreneurial ecosystems. However, this operational emphasis could limit the extent to which the results apply to other forms of generative AI systems (e.g., image-generation or multimodal applications) that entail distinct cognitive and behavioral engagement processes. Future studies could extend the measurement framework by integrating a broader spectrum of AI tools and contrasting user experiences in varied technological settings, thus offering deeper insights into how diverse forms of GAA shape entrepreneurial cognition, emotion, and behavior.

Finally, although cross-sectional survey data provide valuable associative insights, they cannot establish temporal precedence or eliminate alternative causal explanations. Future studies should consider longitudinal or panel designs, or employ experience-sampling methods to capture how psychological responses and SEI evolve as individuals engage with generative AI in dynamic learning and entrepreneurial environments.

## Figures and Tables

**Figure 1 behavsci-15-01705-f001:**
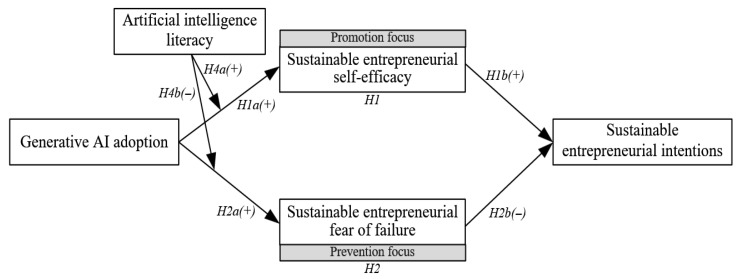
Conceptual model.

**Figure 2 behavsci-15-01705-f002:**
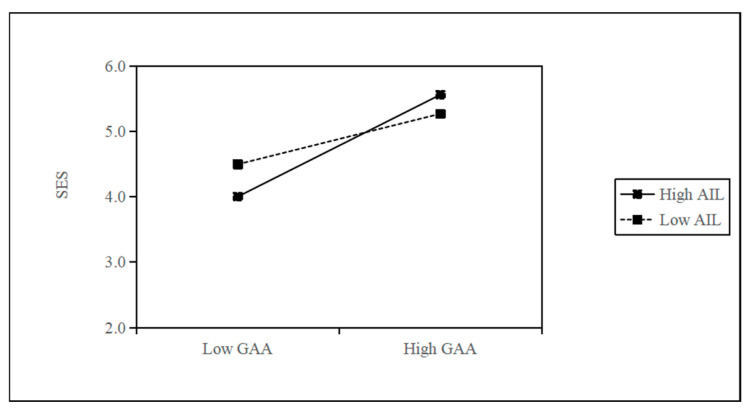
The moderating effect of AIL on GAA and SES.

**Figure 3 behavsci-15-01705-f003:**
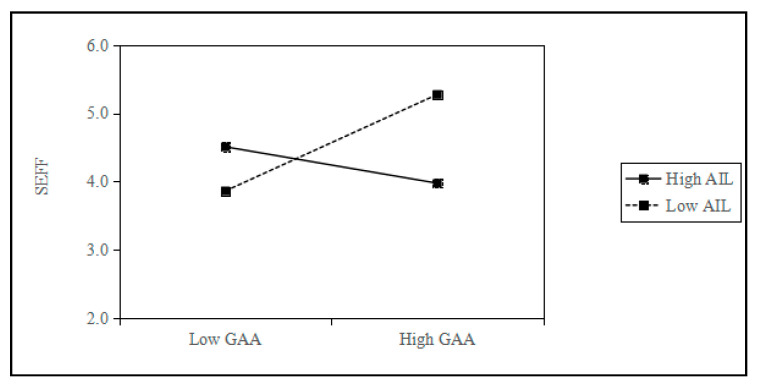
The moderating effect of AIL on GAA and SEFF.

**Figure 4 behavsci-15-01705-f004:**
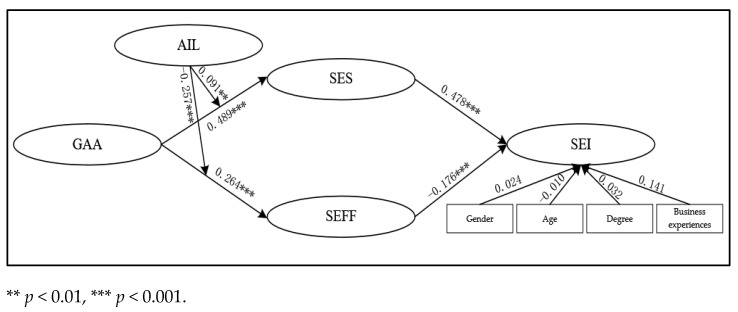
Structural equation model.

**Table 1 behavsci-15-01705-t001:** Relevant studies on AI and sustainable entrepreneurship.

Subject	Evaluation	Antecedents	Outcome	Theory	Main Findings	Reference
Gen AI	Positive	Gen AI Adoption; Perceived AI capacities	Sustainability- oriented entrepreneurial intentions	SOR	GAA has a positive impact on SOI.	[17] ([17])
		ChatGPT adoption	Digital entrepreneurial intentions	EEM	ChatGPT adoption increases digital entrepreneurship intentions and behavior.	[18] ([18])
		Gen AI	Sustainable business model innovation	ABC	Gen AI adoption significantly enhances both exploitative learning and exploratory learning, which in turn drive SBMI.	[60] ([60])
		Gen AI	Entrepreneurial competencies	/	ChatGPT has the potential to improve various dimensions of students’ entrepreneurial skills and capabilities.	[53] ([53])
AI	Positive	Use of AI in teaching	Sustainable entrepreneurial intention	TPB	In entrepreneurial education, integrating AI technology strengthens the link between cognitive precursors and SEI by facilitating hands-on learning experiences and lowering perceived obstacles.	[4] ([4])
		AI tools	Entrepreneurial intentions	Technological acceptance model and TPB	AI functions as a versatile and powerful instructional resource that significantly influences the formation of EI.	[65] ([65])
		AI	Sustainable entrepreneurship	Review	Artificial intelligence positively influences environmental progress within the realm of sustainable entrepreneurship.	[22] ([22])
		AI	Sustainable entrepreneurship	Review	AI serves as a pivotal catalyst in promoting sustainable entrepreneurial initiatives.	[2] ([2])
		AI; Big Data	Sustainable entrepreneurship	Review	AI and BD technologies contribute effectively to incremental sustainability improvements and hold substantial potential for attaining the broader vision of strong sustainability.	[8] ([8])
	Negative	AI technology	Sustainable progress	Review	Ethical concerns surrounding AI technologies often evoke feelings of apprehension among individuals, undermining their trust in AI and consequently obstructing its sustainable development.	[55] ([55])
		AI innovation	Sustainable development	Review and case studies	When overseeing AI innovation aimed at sustainable development, a paradox emerges between generating sustainable value and simultaneously causing its destruction, creating a fundamental tension in management practices.	[38] ([38])

**Table 2 behavsci-15-01705-t002:** Summary of sample characteristics (N = 357).

Characteristic	Item	N	%
Gender	Male	176	49.30
	Female	181	50.70
Age	18–19	48	13.45
	20–21	78	21.85
	22–23	78	21.85
	>23	153	42.86
Degree	Bachelor	204	57.14
	Master	84	23.53
	Doctoral	69	19.33
Business experiences	Yes	83	23.25
	No	274	76.75
University type	Research-oriented	166	46.50
	Teaching-oriented	191	53.50
Geographical region	eastern	132	36.97
	central	135	37.82
	western	90	25.21

**Table 3 behavsci-15-01705-t003:** Result of the unmeasured latent method factor test.

Model	χ^2^	df	χ^2^/df	CFI	TLI	IFI	RMESA
Five-factor model	418.995	395	1.061	0.996	0.995	0.996	0.013
Four-factor model	1157.626	399	2.901	0.862	0.850	0.863	0.073
Three-factor model	1533.160	402	3.814	0.795	0.778	0.796	0.089
Two-factor model	2222.285	404	5.501	0.670	0.645	0.672	0.112
Single-factor model	2995.249	405	7.297	0.537	0.503	0.540	0.133
Model including the five factors and the method factor	373.774	365	1.024	0.998	0.998	0.998	0.008

**Table 4 behavsci-15-01705-t004:** Reliability and convergent validity.

Variables	Items	Factor Loadings	Cronbach’s Alpha	AVE	CR
GAA	GAA1	0.795	0.843	0.519	0.843
	GAA2	0.787			
	GAA3	0.773			
	GAA4	0.770			
	GAA5	0.796			
AIL	AIL1	0.807	0.948	0.603	0.948
	AIL2	0.794			
	AIL3	0.824			
	AIL4	0.796			
	AIL5	0.793			
	AIL6	0.769			
	AIL7	0.850			
	AIL8	0.764			
	AIL9	0.751			
	AIL10	0.837			
	AIL11	0.784			
	AIL12	0.790			
SES	SES1	0.801	0.774	0.536	0.775
	SES2	0.847			
	SES3	0.842			
SEFF	SEFF1	0.801	0.851	0.536	0.852
	SEFF2	0.747			
	SEFF3	0.781			
	SEFF4	0.844			
	SEFF5	0.784			
SEI	SEI1	0.781	0.864	0.560	0.864
	SEI2	0.833			
	SEI3	0.803			
	SEI4	0.809			
	SEI5	0.798			

**Table 5 behavsci-15-01705-t005:** Descriptive statistics and discriminant validity.

Variables	1	2	3	4	5
1. GAA	**0.720**				
2. AIL	0.126 *	**0.776**			
3. SES	0.427 ***	0.021	**0.732**		
4. SEFF	0.199 ***	−0.092	0.149 **	**0.732**	
5. SEI	0.390 ***	0.140 **	0.428 ***	−0.035	**0.748**
M	4.853	4.473	4.857	4.342	4.444
SD	1.364	1.569	1.409	1.520	1.554
VIF	1.372	1.042	1.381	1.084	1.355

**Notes:** Diagonal entries (in bold) are the square root of the AVE (average variances extracted). Entries below the diagonal are correlations. * *p* < 0.05, ** *p* < 0.01, *** *p* < 0.001.

**Table 6 behavsci-15-01705-t006:** HTMT ratio of correlations.

Variables	1	2	3	4	5
1. GAA	-				
2. AIL	0.141	-			
3. SES	0.528	0.052	-		
4. SEFF	0.236	0.104	0.185	-	
5. SEI	0.456	0.154	0.524	0.062	-

**Table 7 behavsci-15-01705-t007:** Regression analysis of mediating effects.

Variables	SES		SEFF		SEI				
	M1	M2	M3	M4	M5	M6	M7	M8	M9
Constant	4.100 *** (14.278)	2.376 *** (7.021)	3.394 *** (10.970)	2.659 *** (6.774)	3.926 *** (12.230)	2.139 *** (5.580)	1.311 *** (3.362)	2.504 *** (6.197)	1.697 *** (4.191)
Gender	0.007 (0.022)	0.023 (0.084)	−0.372 (−1.163)	−0.365 (−1.154)	0.066 (0.198)	0.083 (0.268)	0.075 (0.254)	0.033 (0.106)	0.019 (0.065)
Age	0.059 (0.312)	0.091 (0.524)	0.062 (0.306)	0.076 (0.377)	−0.016 (−0.076)	0.017 (0.088)	−0.014 (−0.077)	0.028 (0.142)	−0.004 (−0.020)
Degree	0.234 (1.420)	0.105 (0.693)	0.299 (1.682)	0.244 (1.381)	0.171 (0.930)	0.038 (0.220)	0.001 (0.007)	0.071 (0.417)	0.038 (0.230)
Business experiences	0.262 (0.885)	0.046 (0.167)	0.612 (1.918)	0.520 (1.640)	0.330 (0.997)	0.106 (0.341)	0.090 (0.304)	0.177 (0.575)	0.169 (0.577)
GAA		0.411 *** (8.128)		0.175 ** (2.986)		0.426 *** (7.442)	0.283 *** (4.758)	0.451 *** (7.831)	0.306 *** (5.169)
SES							0.348 *** (6.045)		0.357 *** (6.266)
SEFF								−0.137 ** (−2.660)	−0.153 ** (−3.108)
R2	0.050	0.200	0.053	0.077	0.025	0.157	0.237	0.174	0.258
Adj-R2	0.039	0.189	0.043	0.064	0.013	0.145	0.224	0.160	0.243
F-value	4.633 ***	17.603 ***	4.9513 ***	5.833 ***	2.216	13.122 ***	18.132 ***	12.303 ***	17.306 ***

** *p* < 0.01, *** *p* < 0.001.

**Table 8 behavsci-15-01705-t008:** The results of bootstrap analysis on indirect mediating effects.

Variables		Coef.	S.E.	95%CILL	95%CIUL
Mediator-SES		0.1432	0.0305	0.0856	0.2059
Mediator-SEFF		−0.0241	0.0130	−0.0526	−0.0032
Dual mediators	Total indirect effect	0.1736	0.0334	0.1096	0.2411
	SES	0.1468	0.0311	0.0864	0.2113
	SEFF	−0.0268	0.0130	−0.0553	−0.0051

**Table 9 behavsci-15-01705-t009:** Regression analysis of moderating effects.

Variables	SES		SEFF	
	M11	M12	M13	M14
Constant	2.505 *** (6.513)	4.357 *** (16.468)	3.115 *** (7.027)	3.626 *** (12.367)
Gender	0.045 (0.165)	0.047 (0.171)	−0.286 (−0.903)	−0.290 (−0.960)
Age	0.086 (0.494)	0.057 (0.331)	0.058 (0.289)	0.129 (0.676)
Degree	0.101 (0.659)	0.139 (0.918)	0.226 (1.286)	0.130 (0.772)
Business experiences	0.044 (0.161)	0.076 (0.279)	0.514 (1.628)	0.435 (1.446)
GAA	0.416 *** (8.146)	0.428 *** (8.426)	0.191 ** (3.251)	0.161 ** (2.868)
AIL	−0.031 (−0.705)	−0.032 (−0.753)	−0.109 * (−2.172)	−0.104 * (−2.186)
GAA × AIL		0.092 ** (2.713)		−0.228 *** (−6.060)
R2	0.202	0.218	0.089	0.176
Adj-R2	0.188	0.202	0.073	0.159
F-value	14.731 ***	13.907 ***	5.699 ***	10.629 ***

* *p* < 0.05, ** *p* < 0.01, *** *p* < 0.001.

**Table 10 behavsci-15-01705-t010:** The results of the moderated mediating indirect effects test.

Mediator	Clusters	Coef.	S.E.	95%CILL	95%CIUL	Index of Moderated Mediation	
						Index	95%CI
SES	High AIL	0.2043	0.0450	0.1209	0.2981	0.0328	[0.0038, 0.0714]
	Low AIL	0.1012	0.0363	0.0283	0.1710		
	High-Low intergroup difference	0.1030	0.0538	0.0133	0.2242		
SEFF	High AIL	0.0299	0.0186	0.0007	0.0728	0.0347	[0.0115, 0.0638]
	Low AIL	−0.0791	0.0286	−0.1404	−0.0275		
	High-Low intergroup difference	0.1090	0.0422	0.0342	0.1993		

## Data Availability

The data presented in this study are available from the corresponding author upon reasonable request. Access is restricted to protect participants’ privacy.

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
