# Peer review of "The Double-Edged Sword Effect of Generative AI Adoption on Students’ Sustainable Entrepreneurship Intentions"

_behavsci, 2025, doi:10.3390/bs15121705_

Round 1
Reviewer 1 Report
Comments and Suggestions for Authors
This paper examines sustainable entrepreneurial intentions in the context of Artificial Intelligence (AI) adoption. A study was conducted among Chinese business university students. A significant dual effect of AI adoption is revealed: it has both a strengthening impact through self-efficacy and an undermining effect through fear of failure. The research is very timely, since AI adoption has become a key player in entrepreneurship, making a better understanding of its effects critically important. Furthermore, given the global imperative for sustainable development and the need to accomplish specific sustainability goals, studying the drivers of these intentions is necessary to efficiently design targeted educational interventions.
The introduction draws upon recent scholarly literature to establish the importance of the study and provide a foundational understanding of previous research. This literature review focuses on works assessing entrepreneurial intentions related to AI adoption and analyzes both the positive and negative perspectives offered by Regulatory Focus Theory (RFT).
The methodology is robust, evidenced by the careful selection of established scales, pilot testing, and necessary contextual modifications to the measurement instrument, ensuring its fit with the Chinese cultural context. While the tables and figures are generally well-illustrated and informative, some formatting amendments are required for consistency and clarity. For example, the diagonal figures in Table 5 (representing factor correlations or loadings) should be set in bold (as stated in notes).
The Results section successfully delivers the findings, and the theoretical contributions are well-presented. Highlighting the importance of AI literacy is a strong practical takeaway.
However, the section on educational interventions requires more detail. We need concrete, actionable suggestions beyond general recommendations. For instance, the findings related to fear of failure should lead to specific advice, such as training focused on reframing failure as a learning opportunity and an inevitable part of the entrepreneurial process, rather than a final outcome.
Author Response
Dear Reviewer,
We sincerely appreciate your thoughtful and constructive comments on our manuscript. We are pleased to know that you find the content highly relevant and impactful for advancing research on sustainable entrepreneurial intentions in the context of generative AI adoption.
In response to your valuable suggestions, we have carefully revised the Managerial Implications section to include more detailed and actionable educational interventions. In particular, we emphasized strategies that help students reframe entrepreneurial failure as a learning opportunity and an inevitable part of the entrepreneurial process, rather than a final outcome. In addition, we have refined the presentation of Table 5 by formatting the diagonal figures in bold to enhance clarity and consistency. (For specific details, please see the attachment.)
Thank you again for your positive evaluation and insightful recommendations, which have greatly helped us strengthen both the theoretical depth and practical contribution of our work.

Reviewer 2 Report
Comments and Suggestions for Authors
The manuscript offers a pertinent and timely examination of the dual (positive and negative) psychological mechanisms by which the adoption of generative AI affects students' sustainable entrepreneurial intentions. Situated within Regulatory Focus Theory, the study enhances a burgeoning discourse that connects AI literacy, entrepreneurship, and sustainability. Nonetheless, various theoretical, methodological, and interpretive dimensions necessitate additional refinement to enhance the paper’s scholarly rigour and internal coherence.
First, even though the literature review is long, it often sounds more like a description than an analysis. It should be made clearer how Regulatory Focus Theory and the proposed empirical model are related. The justification for regarding AI literacy as a moderating variable, instead of an antecedent or mediating construct, requires more robust theoretical support. The study makes causal claims that its cross-sectional design can't fully support; therefore, the discussion should focus on associative relationships instead of causal ones.
Second, the methodology section doesn't make it clear what the sample is like or how it was chosen. The paper talks about stratified and purposive sampling, but it doesn't say what the criteria for stratification are or how the respondents are spread out across institutions and regions. Since some survey questions need a deep understanding of generative AI, the authors should explain how they made sure that the people who took the survey had enough knowledge and experience with these kinds of tools. More information should be given about how the scales were adapted and validated. For example, it should be clear whether confirmatory factor analysis (CFA) was used, how translation and cultural adaptation were handled, and whether the latent constructs match exactly those in the original validated instruments. Given the model's intricacy, employing more sophisticated analytical methods like PLS-SEM or covariance-based SEM could enhance the empirical evidence.
Third, the instruments used to measure seem to be only for certain generative AI applications, like ChatGPT and DeepSeek, which could make them less useful in other situations. The authors must elucidate this operational decision and examine its ramifications. Furthermore, the handling of reverse-coded items and the evaluation of data normality must be clearly documented. Even though the indicators of reliability and validity meet standard levels, it would be better if there was more information about how these results were reached.
Fourth, the results section is mostly clear, but the moderation analysis could use a more in-depth look at what AI literacy means in real life at different levels. The discussion section extrapolates a series of theoretical axioms and managerial implications that exceed what is empirically demonstrated in the data. The conclusions ought to be amended to ensure they are thoroughly anchored in the statistical evidence provided.
Lastly, the English writing is good, but it could be better if it were shorter and clearer. Some sentences are too long, and transitions are used too much, which makes it harder to read. The paper would be more powerful if the phrasing and style were tighter. This study is promising and original, but it needs a lot of work to make its methods clearer, its theoretical basis stronger, and its empirical claims more convincing.
Author Response
Dear reviewer,
We sincerely thank the reviewer for the thoughtful and constructive feedback, which has greatly helped us enhance the overall quality, coherence, and clarity of the manuscript. In response to these insightful suggestions, we have made substantial revisions to the theoretical framework, methodological transparency, and empirical interpretation.
Specifically, we have (1) strengthened the theoretical articulation of Regulatory Focus Theory and clarified how it underpins our dual-pathway model; (2) refined the sampling description and analytical procedures to improve methodological rigor; (3) elaborated on the real-world implications of AI literacy in the moderation analysis; and (4) ensured that the Discussion and Conclusion sections are firmly anchored in empirical evidence, with theoretical implications and educational interventions that are both actionable and contextually meaningful. Furthermore, we have revised the entire manuscript for conciseness and clarity by simplifying long and complex sentences, reducing redundant transitions, and enhancing overall readability. (For specific details, please see the attachment.)
Once again, we sincerely thank the reviewer for the valuable insights and constructive guidance, which have been instrumental in improving the academic rigor and practical relevance of this study.
